# IFR-Explore: Learning Inter-object Functional Relationships in 3D Indoor Scenes

**Qi Li[1,*], Kaichun Mo[2,*], Yanchao Yang[2], Hang Zhao[1], Leonidas Guibas[2]**
[1]Tsinghua University
[2]Stanford University
{liqi17thu, zhaohang0124}@gmail.com
{kaichun, yanchaoy, guibas}@cs.stanford.edu

## Abstract

Building embodied intelligent agents that can interact with 3D indoor environments has received increasing research attention in recent years. While most works focus on single-object or agent-object visual functionality and affordances, our work proposes to study a new kind of visual relationship that is also important to perceive and model – inter-object functional relationships (*e.g.*, a switch on the wall turns on or off the light, a remote control operates the TV). Humans often spend little or no effort to infer these relationships, even when entering a new room, by using our strong prior knowledge (*e.g.*, we know that buttons control electrical devices) or using only a few exploratory interactions in cases of uncertainty (*e.g.*, multiple switches and lights in the same room). In this paper, we take the first step in building AI system learning inter-object functional relationships in 3D indoor environments with key technical contributions of modeling prior knowledge by training over large-scale scenes and designing interactive policies for effectively exploring the training scenes and quickly adapting to novel test scenes. We create a new benchmark based on the AI2Thor and PartNet datasets and perform extensive experiments that prove the effectiveness of our proposed method. Results show that our model successfully learns priors and fast-interactive-adaptation strategies for exploring inter-object functional relationships in complex 3D scenes. Several ablation studies further validate the usefulness of each proposed module.

## 1 Introduction

When first-time entering a hotel room in a foreign country, we may need to search among buttons on the wall to find the one that turns on a specific light, figure out how to turn on the faucets in the bathroom and adjust water temperature, or perform several operations on the TV and the remotes to switch to our favorite channel. Many of these may require no effort if we have used the same models of lamp or faucet before, while the others that we are less familiar with will take more time for us to explore and figure out how things work. We humans can quickly adapt to a new indoor environment thanks to our strong prior knowledge about how objects could be functionally related in general human environments (*e.g.*, the room light can be turned on by pressing a button or toggling a switch) and our amazing capability of using only a few interactions while exploring the environment to learn how things work in cases of uncertainty (*e.g.*, multiple lights and buttons in the room). In this work, we investigate how to equip AI systems with the same capabilities of learning inter-object functional relationships in 3D indoor environments.

With the recent research popularity of embodied AI in perception and interaction, many previous works have studied how to perceive, model, and understand object functionality (Kim et al., 2014; Hu et al., 2016; 2018; 2020) and agent-object interaction (Montesano & Lopes, 2009; Do et al., 2018; Nagarajan & Grauman, 2020; Mo et al., 2021a), just to name a few. However, these works mostly investigate single-object/scene or agent-object scenarios. A few past works (Jiang et al., 2012; Sun et al., 2014; Zhu et al., 2015; Mo et al., 2021b; Cheng et al., 2021) have studied inter-object relationships given a pair of objects or for specific tasks (*e.g.*, placement, stacking). We make a first step bringing up research attentions to a new type of visual quantity – inter-object functional relationship, and an interesting yet underexplored challenging problem of how to learn such generic functional relationships in novel 3D indoor environments.

---

*Equal contribution.

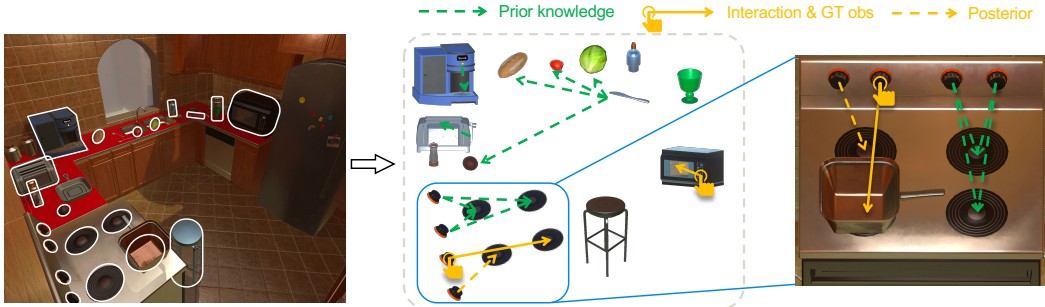

Figure 1: We formulate a novel task of learning inter-object functional relationships (IFRs) in novel 3D indoor scenes. Given an input 3D scene with multiple objects (left), we predict a functional scene graph (middle) with IFRs (*e.g.*, a knife cuts fruits, the microwave button triggers its door open). Our proposed method learns prior knowledge over object geometry (green dashed lines) and posteriors (yellow dashed lines) after performing several interactions (yellow hand signs). In cases of relationship uncertainty such as the stove example (right), learning from the interaction is necessary.

Given a novel indoor scene with multiple objects as the input (Fig. 1, left), we predict a complete directional graph of inter-object functional relationships (Fig. 1, middle) on how the state change of one object causes functional effects on other objects. While the semantics of individual objects already suggest some functional priors (*e.g.*, buttons can be pressed, electrical devices can be turned on), our system learns to build up the correspondence among them (*e.g.*, which button controls which electrical device). Also, since one may encounter unknown object categories exploring novel scenes, we do not assume semantic labels of objects and only use shape geometry as input. Our system, therefore, learns inter-object functional geometry-mappings to what to trigger (*e.g.*, one button among others on the wall) and what other shapes the interaction effects (*e.g.*, one light in the room).

Though seemingly simple as one may interact with all objects in the scene to query the functional scene graph, it is actually very challenging how to efficiently and accurately perceive, reason, and explore the scene with as few interactions as possible. First of all, we do not want to waste time interacting with books and bags as they will not functionally trigger other objects. Secondly, the semantics of two objects may directly indicate functional relationships. For example, we know that buttons control electrical devices in general. Besides, in cases of having two buttons and two lights in the scene, one may guess the pairings according to spatial proximity. All of these examples indicate that learning some geometric and semantic priors among the objects may help predict many inter-object functional relationships even without any interaction. However, there are always scenarios where uncertainties exist. For example, for the stove highlighted in Fig. 1 (right), interactions have to be taken place to figure out which knob controls which burner, though only two interactions are needed for the four pairs if our agent can smartly infer the other two automatically.

We design a two-staged approach, under a self-supervised interactive learning-from-exploration framework, to tackle the problem. The first stage perceives and reasons over the objects in a novel test scene to predict possible inter-object functional relationships using learned functional priors about the object geometry and scene layout. Then, in the second stage, our system performs a few interactions in the scene to reduce uncertainties and fastly adapts to a posterior functional scene graph prediction. We jointly train a reinforcement learning policy to explore large-scale training scenes and use the collected interaction observations to supervise the prior/posterior networks in the system.

Since we are the first to explore this task to our best knowledge, we create a new hybrid dataset based on AI2THOR (Kolve et al., 2017) and PartNet (Mo et al., 2019b) as the benchmark, on which we conduct extensive experiments comparing to various baselines and illustrating our interesting findings. Experiments validate that both stages contribute to the final predictions and prove the training synergy of the exploration policy and the prior/posterior networks. We also find reasonable generalization capabilities when testing over novel scene types with unknown object categories.

In summary, this work makes the following main contributions:
- we first formulate an important, interesting, yet underexplored challenging task – learning inter-object functional relationships in novel 3D indoor environments;
- we propose a novel self-supervised learning-from-exploration framework combining a prior-inference stage and a fast-interactive-adaptation stage to tackle the problem;

- we create a new dataset for benchmarking the proposed task and demonstrating the effectiveness of the proposed approach.

## 2 RELATED WORKS

**Visual Functionality and Affordance.**    We humans perceive and interact with the surrounding 3D world to accomplish everyday tasks. To equip AI systems with the same capabilities, researchers have done many works investigating shape functionality (Kim et al., 2014; Hu et al., 2016; 2018; 2020; Lai et al., 2021; Guan et al., 2020), grasp affordance (Lenz et al., 2015; Redmon & Angelova, 2015; Montesano & Lopes, 2009; Qin et al., 2020; Kokic et al., 2020; Mandikal & Grauman, 2021; Yang et al., 2020; Corona et al., 2020; Kjellström et al., 2011; Fang et al., 2018; Nagarajan et al., 2019; Brahmbhatt et al., 2019; Jiang et al., 2021), manipulation affordance (Do et al., 2018; Nagarajan et al., 2019; 2020; Nagarajan & Grauman, 2020; Mo et al., 2021a), scene affordance (Piyathilaka & Kodagoda, 2015; Rhinehart & Kitani, 2016; Li et al., 2019b), etc. While these works mostly study single-object/scene or agent-object interaction scenarios, our work explores inter-object relationships. Besides, we investigate how to explore functional relationships by interacting in 3D scenes, different from previous works that study interactions given specific object pairs or tasks (Jiang et al., 2012; Sun et al., 2014; Zhu et al., 2015; Mo et al., 2021b; Cheng et al., 2021).

**Inter-object Relationships.**    Besides perceiving individual objects in the environment, understanding the rich relationships among them is also crucial for many downstream applications. Previous works have attempted to model object supporting and contact relationships (Fisher et al., 2011; Silberman et al., 2012), spatial relationships (Galleguillos et al., 2008; Gould et al., 2008; Kulkarni et al., 2019), semantic relationships (Sadeghi & Farhadi, 2011; Divvala et al., 2014), physical relationships (Mitash et al., 2019), etc. More recently, researchers have introduced scene graphs (Liu et al., 2014; Johnson et al., 2015; Krishna et al., 2017; Xu et al., 2017; Li et al., 2017b; Zhu et al., 2020; Wald et al., 2020) and hierarchies (Armeni et al., 2019; Li et al., 2019a; 2017a; Mo et al., 2019b;a) to parse the objects, object parts, and their relationships. Unlike the spatial, geometric, semantic, or physical relationships investigated in these works, we pay more attention to inter-object functional relationships in which the state change of one object causes functional effects on other objects.

**Learning from Exploration and Interaction.**    While deep learning has demonstrated great success in various application domains (Russakovsky et al., 2015; Silver et al., 2016), large-scale annotated data for supervision inevitably becomes the bottleneck. Many works thus explore self-supervised learning via active perception (Wilkes & Tsotsos, 1992), interactive perception (Bohg et al., 2017), or interactive exploration (Wyatt et al., 2011) to learn visual representations (Jayaraman & Grauman, 2018; Yang et al., 2019; Weihs et al., 2019; Fang et al., 2020; Zakka et al., 2020), objects and poses (Caicedo & Lazebnik, 2015; Han et al., 2019; Deng et al., 2020; Chaplot et al., 2020b; Choi et al., 2021), segmentation and parts (Katz & Brock, 2008; Kenney et al., 2009; Van Hoof et al., 2014; Pathak et al., 2018; Eitel et al., 2019; Lohmann et al., 2020; Gadre et al., 2021), physics and dynamics (Wu et al., 2015; Mottaghi et al., 2016; Agrawal et al., 2016; Li et al., 2016; Janner et al., 2018; Xu et al., 2019; Lohmann et al., 2020; Ehsani et al., 2020), manipulation skills (Pinto & Gupta, 2016; Levine et al., 2018; Agrawal et al., 2016; Pinto et al., 2016; Pinto & Gupta, 2017; Zeng et al., 2018; Batra et al., 2020), navigation policies (Anderson et al., 2018; Chen et al., 2019; Chaplot et al., 2020a; Ramakrishnan et al., 2021), etc. In this work, we design interactive policies to explore novel 3D indoor rooms and learn our newly proposed inter-object functional relationships.

## 3 PROBLEM FORMULATION

We formulate a novel problem of inferring inter-object functional relationships (IFRs) among objects in novel 3D indoor environments (Fig. 1). For the task input, each scene $S$ comprises multiple 3D objects $\{O_1, O_2, \cdots, O_n\}$, each of which is presented as a 3D point cloud with 2,048 points depicting the complete shape geometry. Every object $O_i$ may be subject to state changes (*e.g.*, water is running or not from a faucet, a button is pressed up/down). The set of 3D objects describe the complete 3D scene $S = \{O_1, O_2, \cdots, O_n\}$ including both the objects with and without possible state changes. The spatial layout of the objects in the scene is explicitly modeled that each object $O_i = (\hat{O}_i, c_i, s_i)$ can be decomposed into a zero-centered unit-sized shape point cloud $\hat{O}_i \in \mathbb{R}^{2,048 \times 3}$, the object center position in the scene $c_i \in \mathbb{R}^3$, and its isotropic scale $s_i \in \mathbb{R}$. For the task output, we are interested to figure out the (IFRs) $\mathscr{R}_S$. In a case that interacting with an object $O_i$ (a *trigger*) will cause a functional state change of another object $O_j$ (a *responder*), we formally define an observed IFR $(O_i, O_j) \in \mathscr{R}_S$. In the task output $\mathscr{R}_S$, there could be one-to-one or many-to-many relationships between two objects, as a trigger object may cause functional state changes of many responder objects and vice versa.

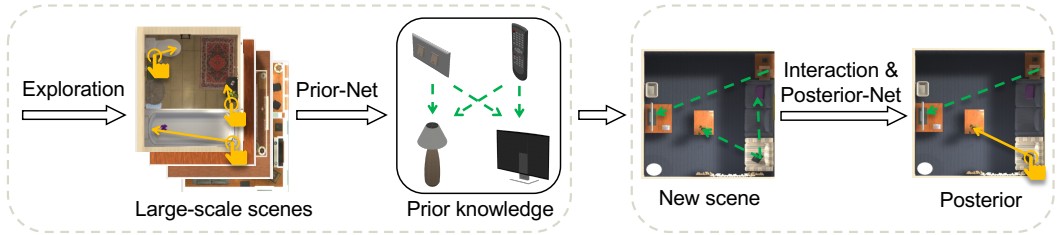

Figure 2: **System Overview.** During training (left), we jointly learn an RL policy to explore large-scale scenes and use the collected interaction observations to supervise the prior/posterior networks. In the test time (right), given a novel scene, we first propose possible inter-object functional relationships by applying the learned prior knowledge and then perform interactions to resolve uncertain cases.

We define a task scenario that an agent actively explores the environment and interacts with the objects to output the inter-object functional relationships following the popular embodied and real-world settings (Nagarajan & Grauman, 2020; Ramakrishnan et al., 2021). For each interaction, an agent picks an object, triggers its state change, and directly receives the state changes of the other objects that have been affected in the scene. We assume that standardized visual quantities (*e.g.*, object segmentation, state changes) are provided to the agent as they are well-established research topics by themselves (Geiger et al., 2012; Li et al., 2020), and thus that the functional effect of a relationship is directly observable. An agent is provided with large-scale scenes to explore for learning in the training stage and is asked to predict the functional scene graph $(S, \mathscr{R}_S)$ for a novel scene at the test time. We also abstract away complexities on robotic navigation (Anderson et al., 2018; Ramakrishnan et al., 2021) and manipulation (Nagarajan & Grauman, 2020; Mo et al., 2021a), which are orthogonal to our contribution to estimating inter-object functional relationships.

## 4 TECHNICAL APPROACH

Fig. 2 presents the system overview. Given a novel indoor scene, the prior-inference stage first produces a functional scene graph prior $(S, \mathscr{R}_S^s)$ by perceiving and reasoning over the objects in the scene $S = \{O_1, O_2, \cdots, O_n\}$. Then, in the fast-adaptation stage, the agent performs as few interactions as possible to reduce uncertainties and updates prior into posterior for the final functional scene graph prediction $(S, \mathscr{R}_S)$. We train an exploration policy that learns to effectively and efficiently explore large-scale training scenes and collects interaction data to supervise the prior/posterior networks.

### 4.1 THE PRIOR-INFERENCE STAGE

We design two prior networks – BR-Prior-Net and SR-Prior-Net, for the functional scene graph prior learning, where BR-Prior-Net focus on modeling binary relationship priors $r_{i,j}^b \in [0,1]$ given only two input shape geometry $(O_i, O_j)$ ($\forall i, j$) while SR-Prior-Net reasons scene-level functional relationship priors $\mathscr{R}_S^s = \{r_{i,j}^s \in [0,1] | O_i, O_j \in S\}$ considering all the objects in the scene $S = \{O_1, O_2, \cdots, O_n\}$.

#### 4.1.1 BINARY-RELATIONSHIP PRIOR NETWORK (BR-PRIOR-NET)

Given two objects in the scene (*e.g.*, a button and a light), humans often have binary priors if the two objects are functionally correlated or not (*e.g.*, the button may control the light) according to geometry and semantics of the two shapes, regardless of their locations in the scene. Having this observation, we thus design a binary-relationship prior network (BR-Prior-Net) to model the possibility of functional relationship for every pair of trigger object $O_i$ and responder object $O_j$.

As shown in Fig. 3 (left), our BR-Prior-Net takes two object point clouds $O_i, O_j \in \mathbb{R}^{2048}$ as input and outputs the belief of IFR $r_{i,j}^b \in [0,1]$. By allowing $O_i = O_j$, we also consider self-relationship (*e.g.*, triggering the button on the microwave opens its door). We train a shared PointNet++ network Qi et al. (2017), with four set abstraction layers with $N = [512, 128, 32, 1]$ point resolutions, to process the two point clouds and output global features $f_i^b, f_j^b \in \mathbb{R}^{64}$ that summarize the two input shapes. We then concatenate the two obtained features and employ a Multilayer Perceptron (MLP) network to estimate $r_{i,j}^b \in [0,1]$ for the final output. We obtain the binary functional scene relationship prior $\mathscr{R}_S^b = \{r_{i,j}^b \in [0,1] | O_i, O_j \in S\}$ by enumerating all object pairs in the scene.

#### 4.1.2 SCENE-RELATIONSHIP PRIOR NETWORK (SR-PRIOR-NET)

While the binary-relationship prior network empirically performs well in predicting pairwise functional relationships as it effectively scopes down the problem scale to two input objects only, much

Figure 3: **Network Architecture.** The BR-Prior-Net takes two object point clouds as input and estimates their functional relationship likelihood. By passing all object pairs through BR-Prior-Net, we obtain a binary functional graph prior $\mathscr{R}_S^b$. Then, we employ a Graph Convolutional Network (GCN) to model the scene context and predict a scene functional graph prior $\mathscr{R}_S^s$. We repurpose the same GCN as SR-Posterior-Net which produces a scene posterior graph $\mathscr{R}_S^{(t)}$ after each interaction step $t$. The agent outputs a final functional scene graph prediction $\mathscr{R}_S$ after addressing all uncertainties.

important scene-level information is ignored, such as relative object poses and scene object layout. Such scene context may provide clues on relationship pairings in cases of uncertainties. For example, the BR-Prior-Net may connect all buttons to all lights in the room, but given the object layout, one may assign each button to its closest light with higher probabilities. Therefore, we further introduce scene-relationship prior network (SR-Prior-Net) to take the scene context into consideration.

The SR-Prior-Net is implemented as a three-layer Graph Convolutional Network (GCN) (Kipf & Welling, 2016) that connects and passes messages among all objects in the scene. Fig. 3 (right) shows an overview of the design. Each object $O_i$ is represented as a node $i$ in the graph, with a node feature $n_i^{(0)}$ that combines the PointNet++ geometry feature and the scene-contextual information (*e.g.*, object position and size). The edge between two nodes is initialized with $w_{i,j}^{(0)} = r_{i,j}^b$ predicted by the BR-Prior-Net. To allow better graph message passing throughout the graph, we additionally connect two nodes if the two objects are close enough ($< 0.5m$, which covers about 53.1% IFRs in our dataset) because closer objects are more likely to have functional relationships, *e.g.*, a stove knob is close to the corresponding stove burner. We use an edge weight of $w_{i,j}^{(0)} = \gamma = 0.6$, which is experimentally tuned out (see Table E.5). After three iterations of graph convolutions

$$n_i^{(k)} = \Theta \sum_j \frac{w_{j,i}^{(k-1)}}{\sqrt{d_j^{(k-1)} d_i^{(k-1)}}} n_j^{(k-1)}, \forall k = 1, 2, 3 \tag{1}$$

where $d_i^{(k-1)} = 1 + \sum_j w_{j,i}^{(k-1)}$ and $\Theta$ denotes the learnable parameters, the network outputs a scene-contextual embedding $f_i^S = n_i^{(3)} \in \mathbb{R}^{32}$ for each object $O_i$. Next, for each object pair, we concatenate their feature embeddings, feed it through an MLP, and predict a likelihood score for their functional relationship $r_{i,j}^s \in [0,1]$. The final scene-level functional scene relationship prior $\mathscr{R}_S^s = \{r_{i,j}^s \in [0,1] | O_i, O_j \in S\}$ is obtained by enumerating all object pairs in the scene.

### 4.2 THE FAST-INTERACTIVE-ADAPTATION STAGE

While the prior networks do their best to propose functionally related objects by perceiving and reasoning over objects in the scene, there are always uncertain cases that priors do not suffice (*e.g.*, the stove example in Fig. 1 (right)), where interactions are needed to reduce such uncertainties. A smart agent only needs to perform very few interactions as needed and is capable of inferring other pairings. We therefore design SR-Posterior-Net to learn such posterior reasoning after interactions and propose a test-time fast-adaptation strategy that sequentially picks a few objects to interact.

#### 4.2.1 SCENE-RELATIONSHIP POSTERIOR NETWORK (SR-POSTERIOR-NET)

The scene-relationship posterior network (SR-Posterior-Net) shares the same network architecture and weights to SR-Prior-Net introduced in Sec. 4.1.2. We only repurpose the network to model functional scene graph posteriors by simply changing the network input to the current functional scene relationship belief modulated by the observed interaction outcomes. In our experiments, we perform interactions over objects one-by-one sequentially and thus feed through SR-Posterior-Net multiple times to evolve the functional scene relationship posteriors $\mathscr{R}_S^{(t)}$ ($t = 1, 2, \cdots, T_S$) after each interaction, where $T_S$ denotes the total number of interactions performed in this scene.

Concretely, when $t = 0$, we use $\mathscr{R}_S^{(0)} = \mathscr{R}_S^s$, which is the functional scene relationship prior estimated by SR-Prior-Net without performing any interaction. For each timestep $t$, given the interaction observations $\{e_{i_t,j} \in \{0,1\} | O_j \in S\}$ after interacting with object $O_{i_t}$, where $e_{i_t,j} = 1$ if and only if $O_{i_t}$ triggers $O_j$, we update the posterior $\mathscr{R}_S^{(t)}$ with all past observations $r_{i,j}^{(t)} = e_{i,j}$ ($i = i_1, i_2, \cdots, i_t, j = 1, 2, \cdots, n$), feed the modified posterior through SR-Posterior-Net, and obtain a functional scene relationship posterior $\mathscr{R}_S^{(t+1)}$ for the next timestep $t + 1$. We compute the final functional scene relationship prediction $\mathscr{R}_S = \{(O_i, O_j) | r_{i,j}^{(T_S)} \in \mathscr{R}_S^{(T_S)} > \tau, \forall i, j\}$ with a threshold $\tau = 0.9$.

### 4.2.2 TEST-TIME FAST-INTERACTIVE-ADAPTATION STRATEGY

Now, the only questions left are how to pick the object to interact with at each timestep and when to stop. We find that a simple heuristic to explore the object that is predicted most uncertainly in our posterior works very well in our experiments. Specifically, at each step $t$, the agent picks $O_i$ that

$$i = \arg\max_i \left( \max_j \left( \min \left( r_{i,j}^{(t)}, 1 - r_{i,j}^{(t)} \right) \right) \right), \forall t = 0, 1, \cdots, T_S - 1 \tag{2}$$

We stop the test-time fast-interactive-adaptation procedure when all predictions are certain enough,

$$\min \left( r_{i,j}^{(t)}, 1 - r_{i,j}^{(t)} \right) < \gamma, \forall i, j \tag{3}$$

with a threshold $\gamma = 0.05$, the agent explores about 19% of objects in the room on average. $\gamma$ can also be enlarged to reduce number of interactions. Another strategy is to fix the test-time interaction budget, *e.g.*, 10% or 20% of the total number of objects in each room.

## 4.3 SELF-SUPERVISED LEARNING FROM EXPLORATION

We adopt the embodied paradigm of learning from exploration and interaction (Nagarajan & Grauman, 2020; Ramakrishnan et al., 2021) that an agent actively explores the large-scale training scenes, interacts with the environments, and collects data for supervision. It is a challenging task how to effectively probe the environments for self-supervised learning and perform as few interactions as possible for efficiency. We formulate the task as a reinforcement learning (RL) problem and learn an exploration policy that collects data for supervising the prior and posterior networks.

### 4.3.1 EXPLORATION POLICY

The RL policy, implemented under the Proximal Policy Optimization (PPO) (Schulman et al., 2017) framework, takes the current functional scene graph representation as the state input $state^{(t)} = (S, \mathscr{R}_S^{(t)})$ at each timestep $t$ and picks one object as the action output to interact with $action^{(t)} = O_i \in S$. When $t = 0$, we use $\mathscr{R}_S^{(0)} = \mathscr{R}_S^s$, which is predicted by the SR-Prior-Net (Sec. 4.1.2). After each interaction, the agent observes the interaction outcome that some or no objects have functional state changes triggered by $O_i$, namely $observation^{(t)} = \{e_{i,j} \in \{0,1\} | O_j \in S\}$ where $e_{i,j} = 1$ if and only if $O_i$ triggers $O_j$, which will then be passed to the SR-Posterior-Net (Sec. 4.2.1) to update the functional relationship belief $\mathscr{R}_S^{(t+1)}$ for the next timestep $t + 1$. The agent may decide to stop exploring the current scene and move on to the next one under a limited interaction budget.

We design the RL reward function as follows.

$$reward^{(t)} = \alpha \max_j |r_{i,j}^{(t)} - e_{i,j}| + \beta \mathbb{1}[e_{i,j}] - \gamma \tag{4}$$

The first term encourages interacting with $O_i$ to correct the wrong or uncertain functional relationship beliefs $r_{i,j}^{(t)}, \forall j$. There are three cases for this term: 1) the agent's belief is already correct ($r_{i,j}^{(t)} = e_{i,j}$, thus $|r_{i,j}^{(t)} - e_{i,j}| = 0$); 2) the belief is wrong ($r_{i,j}^{(t)} = 1 - e_{i,j}$, thus $|r_{i,j}^{(t)} - e_{i,j}| = 1$); 3) the agent is uncertain (*e.g.*, $r_{i,j}^{(t)} = 0.5$, thus $|r_{i,j}^{(t)} - e_{i,j}| = 0.5$). We use *max* instead of *avg* as the aggregation function because the IFRs are sparse and an average value may conceal the mistake. The second term rewards exploring trigger objects of interest, *e.g.*, it encourages the agent to interact with buttons or switches, instead of boxes or cabinets. The third term simulates the interaction cost to avoid redundant interactions for efficient exploration. We use $\alpha = 2$, $\beta = 1$ and $\gamma = 1$ in our experiments.

The RL network shares the same three-layer GCN backbone as the scene-relationship networks (Sec. 4.1.2 and 4.2.1) for reasoning over the functional scene graph $(S, \mathscr{R}_S^{(t)})$. However, instead of predicting IFRs in SR-Prior-Net (Sec. 4.1.2) and SR-Posterior-Net (Sec. 4.2.1), the GCN RL policy

outputs action probabilities $a_i^{(t)}$'s over object nodes $O_i$'s, by firstly obtaining an embedding $n_i^{(t)} \in \mathbb{R}^{32}$ for each object $O_i$, then concatenating with a global scene feature $n_S^{(t)}$ computed by averaging across all object embeddings, and finally feeding through an MLP to get an action score $a_i^{(t)}$. In addition, we duplicate the scene feature $n_S^{(t)}$ and input it to the MLP to query the termination score $a_{stop}$. All the action scores are finally passed through a SoftMax layer for the action probability scores.

### 4.3.2 TRAINING STRATEGY AND LOSSES

To supervise the proposed interactive learning-from-exploration framework, we alternate the training of the exploration RL policy and the prior/posterior networks. We leverage the training synergy that the prior/posterior networks predict helpful information for better exploration of the scenes, while the exploration RL policy collects useful data to better train the two networks. In our implementation, we train the exploration policy and the prior/posterior networks to make they jointly converge.

In each loop, conditioned on the current prior/posterior predictions, the exploration policy strategically explores the large-scale training scenes with a total interaction budget of 1000 (*e.g.*, roughly 32% of the total 3125 objects from 210 scenes). The exploration policy learns to wisely allocate budgets across different scenes and pick the objects worth interacting with. Specifically, the agent can interact with $m$ objects in one scene and explore the rest scenes with the $1000 - m$ budget left. For a scene $S = \{O_1, O_2, \cdots, O_n\}$ where the agent sequentially performs $m$ interactions over objects $(O_{i_1}, O_{i_2}, \cdots, O_{i_m})$, we obtain $m$ sets of observations $(\{e_{i_1,j} \in \{0,1\}|O_j \in S\}, \{e_{i_2,j} \in \{0,1\}|O_j \in S\}, \cdots, \{e_{i_m,j} \in \{0,1\}|O_j \in S\})$.

We then use the collected interaction observations to train the prior/posterior networks. For BR-Prior-Net, we simply train for the observed link predictions $r_{i,j}^b \to e_{i,j}$ $(i = i_1, i_2, \cdots, i_m, j = 1, 2, \cdots, n)$. We learn one shared network for SR-Prior-Net and SR-Posterior-Net but train it with two supervision sources. For learning the SR-Priors, we input the BR-Prior-Net prediction $\mathscr{R}_S^b$, output the scene-level prior $\mathscr{R}_S^s$ and train for the observed link predictions as well $r_{i,j}^s \to e_{i,j}$ $(i = i_1, i_2, \cdots, i_m, j = 1, 2, \cdots, n)$.

For training the SR-Posteriors, we use the posterior estimate $\mathscr{R}_S^{(t)}$ at each timestep $t = 1, 2, \cdots, m - 1$, update it with the past ground-truth observations $r_{i,j}^{(t)} = e_{i,j}$ $(i = i_1, i_2, \cdots, i_t, j = 1, 2, \cdots, n)$, input to the network to obtain the posterior estimate $\mathscr{R}_S^{(t+1)}$, and supervise the link predictions $r_{i,j}^{(t+1)} \to e_{i,j}$ $(i = i_{t+1}, i_{t+2}, \cdots, i_m, j = 1, 2, \cdots, n)$. We use the standard binary cross entropy for all loss terms.

## 5 EXPERIMENTS

We evaluate our method on a new hybrid dataset and compare it to several baselines that prove the effectiveness of the proposed method and its components. We also experiment with transferring to novel scene types that may contain novel shape categories. See appendix for more results.

### 5.1 DATASET

We create a new hybrid dataset based on AI2THOR (Kolve et al., 2017) and PartNet-Mobility (Mo et al., 2019b; Xiang et al., 2020) to support our study. AI2THOR provides 120 interactive scenes of 4 types (*i.e.*, bathroom, bedroom, living room, and kitchen) containing more than 1500 interactive objects covering 102 categories. The PartNet-Mobility dataset contains 2,346 3D articulated object CAD models from 46 categories. We replace 83 AI2THOR objects with 888 PartNet-Mobility objects for 31 categories (12 exist in AI2THOR while 19 are newly added) to further enrich the diversity of the shape geometry. We use 27 kinds of functional relationships originally built in AI2THOR (*e.g.*, the microwave can be toggled on) and also enrich 21 more types of inter-object functional relationships (*e.g.*, the pen can be used to write in the book).

In total, our dataset contains 1200 scenes covering 23360 objects from 121 categories. We enrich the origin 120 AI2THOR scenes by 1) randomly spawn objects into different locations and 2) randomly replace the object geometries of the same type. Fig. A.6 shows one example room from each of the four types. Among these, 27 are *trigger* objects, 29 are *responders*, where 11 are both *trigger* and *responders* (*e.g.*, desk lamp), and 79 are non-interactive background objects. We split the dataset into non-overlapping 800 training scenes and 400 test scenes. Table A.3 presents more detailed statistics.

### 5.2 BASELINES AND METRICS

Since we are the first to study the problem to our best knowledge, there is no external baseline for directly fair comparisons. However, to validate the effectiveness for each of our proposed

Table 1: **Quantitative Evaluations.** We compare to several baselines or ablated versions, and report the precision (**P**), recall (**R**), F1-score (**F1**), and Matthews correlation coefficient (**MCC**) given the test-time interaction budget as 10% or 20% of the per-room object counts.

| Method | Bathroom | | | | Bedroom | | | | Living Room | | | | Kitchen | | | |
|---|---|---|---|---|---|---|---|---|---|---|---|---|---|---|---|---|
| | P | R | F1 | MCC | P | R | F1 | MCC | P | R | F1 | MCC | P | R | F1 | MCC |
| Random (10%) | **1.000** | 0.108 | 0.195 | 0.317 | **1.000** | 0.109 | 0.197 | 0.258 | **1.000** | 0.102 | 0.185 | 0.257 | **1.000** | 0.126 | 0.224 | 0.282 |
| Random (20%) | **1.000** | 0.232 | 0.377 | 0.438 | **1.000** | 0.201 | 0.335 | 0.427 | **1.000** | 0.216 | 0.355 | 0.427 | **1.000** | 0.209 | 0.346 | 0.402 |
| Abla-NoBinaryPrior (10%) | 0.637 | 0.952 | 0.759 | 0.774 | 0.476 | 0.945 | 0.628 | 0.666 | 0.297 | 0.850 | 0.425 | 0.488 | 0.553 | 0.863 | 0.671 | 0.687 |
| Abla-NoScenePrior (10%) | 1.000 | 0.274 | 0.422 | 0.515 | 0.870 | 0.178 | 0.287 | 0.383 | 0.774 | 0.578 | 0.648 | 0.661 | 0.663 | 0.734 | 0.686 | 0.690 |
| Abla-RandomExplore (10%) | 0.543 | 0.934 | 0.683 | 0.708 | 0.348 | 0.908 | 0.497 | 0.556 | 0.516 | 0.845 | 0.634 | 0.656 | 0.597 | 0.799 | 0.680 | 0.687 |
| Abla-RandomAdapt (10%) | 0.739 | 0.891 | 0.805 | 0.813 | 0.706 | 0.866 | 0.767 | 0.776 | 0.710 | 0.739 | 0.715 | 0.720 | 0.585 | 0.701 | 0.635 | 0.624 |
| Abla-NoBinaryPrior (20%) | 0.724 | 0.979 | 0.828 | 0.838 | 0.841 | 0.980 | 0.897 | 0.903 | 0.617 | **0.940** | 0.715 | 0.743 | 0.700 | 0.921 | 0.788 | 0.798 |
| Abla-NoScenePrior (20%) | **1.000** | 0.530 | 0.677 | 0.717 | 0.980 | 0.449 | 0.598 | 0.650 | 0.851 | 0.733 | 0.776 | 0.783 | 0.783 | 0.855 | 0.809 | 0.813 |
| Abla-RandomExplore (20%) | 0.617 | 0.954 | 0.745 | 0.763 | 0.461 | 0.948 | 0.610 | 0.653 | 0.631 | 0.926 | 0.743 | 0.760 | 0.776 | **0.906** | 0.832 | 0.835 |
| Abla-RandomAdapt (20%) | 0.773 | 0.915 | 0.834 | 0.820 | 0.720 | 0.881 | 0.781 | 0.788 | 0.743 | 0.772 | 0.748 | 0.749 | 0.618 | 0.714 | 0.659 | 0.668 |
| Interaction-Exploration (10%) | **1.000** | 0.327 | 0.482 | 0.561 | 0.681 | 0.594 | 0.618 | 0.626 | 0.952 | 0.477 | 0.620 | 0.663 | 0.950 | 0.176 | 0.282 | 0.385 |
| Interaction-Exploration (20%) | **1.000** | 0.528 | 0.683 | 0.720 | 0.742 | 0.668 | 0.688 | 0.695 | 0.972 | 0.627 | 0.748 | 0.772 | **1.000** | 0.334 | 0.478 | 0.556 |
| Ours-Final (10%) | 0.810 | 0.960 | 0.875 | 0.878 | 0.765 | 0.956 | 0.843 | 0.851 | 0.801 | 0.861 | 0.825 | 0.827 | 0.667 | 0.789 | 0.718 | 0.721 |
| Ours-Final (20%) | 0.937 | **0.984** | **0.957** | **0.958** | 0.877 | **0.987** | **0.924** | **0.927** | 0.892 | 0.924 | **0.903** | **0.905** | 0.848 | 0.890 | **0.864** | **0.865** |

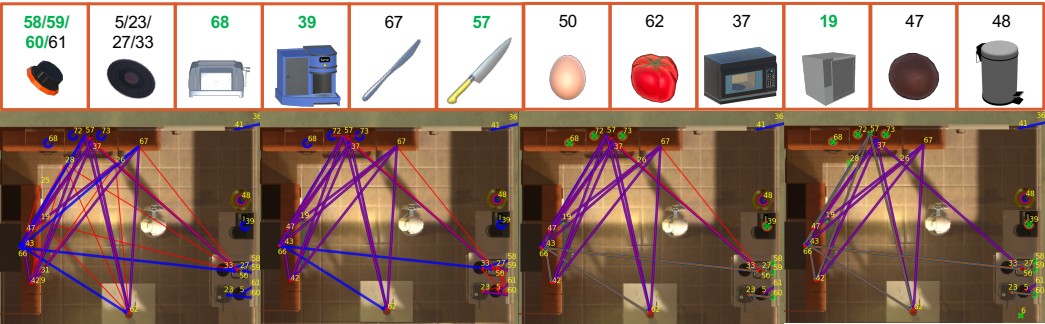

Figure 4: **Qualitative Results.** We show top-down views of one example scene. In the bottom row, from left to right, we show the functional scene-graph predictions given by binary priors, scene priors, posteriors with 10% interactions, posteriors with 20% interactions. In these figures, we mark the ground-truth relationships in blue lines, the predicted ones in red lines, the interaction observations in green lines, and the interacted objects with green cross symbols. We assign each object with a yellow ID linking to one of the top-row subfigures for some example object zoom-ins. The green IDs in the top row mean the object has been interacted during exploration.

modules (*e.g.*, BR-Prior-Net, SR-Prior-Net, the fast-interactive-adaptation strategy, the self-supervised exploration policy), we compare **Ours-Final** to several baselines and ablated versions:

- **Random**: the agent performs random actions in the scene at the test time;
- **Abla-NoBinaryPrior**: the agent does not use the binary priors and instead initializes the input edges to SR-Prior-Net with $w_{i,j}^{(0)} = 1$;
- **Abla-NoScenePrior**: the agent uses the predicted binary-relationship prior $\mathscr{R}_S^b$ instead of the estimated scene-relationship prior $\mathscr{R}_S^s$ as the prior knowledge;
- **Abla-RandomExplore**: the agent does not use the proposed exploration policy and instead randomly explores training scenes during training;
- **Abla-RandomAdapt**: the agent randomly picks objects to interact with during adaptation;
- **Ours-PriorOnly**: the agent directly uses the scene-relationship prior predictions $\mathscr{R}_S^s$ and cuts a threshold 0.5 to predict the final outputs without any interaction;
- **Interaction-Exploration**: we try our best to adapt Nagarajan & Grauman (2020) to our setting, though it originally tackles a very different task on learning affordance over individual shapes in a scene and does not propose IFRs. We use a PointNet++ to encode object features and an MLP to additionally predict a set of objects that each interaction at a object causes.

We use the standard precision, recall, and F1-score for quantitative evaluations and comparisons. Additionally, we set a per-room interaction budget for fairly comparing all the methods and report their performance when the budgets are 10% and 20% of the object count in each room.

## 5.3 RESULTS AND ANALYSIS

Table 1 presents the quantitative evaluations and comparisons to baselines, where we see that **Ours-Final** achieves the best scores in F1-score and MCC in all comparisons under different interaction budgets. The **Random** baseline observes the direct interaction outcomes and thus by design gives a

Table 2: We present quantitative evaluations when transferring our network trained on one type of rooms to other types at the test time that may also contain unseen object types. We additionally show results transferring to RoboTHOR, whose scenes are visually different from AI2THOR.

| Method | Bathroom | | | Bedroom | | | Living Room | | | Kitchen | | | Robo | | |
|---|---|---|---|---|---|---|---|---|---|---|---|---|---|---|---|
| | P | R | F1 | P | R | F1 | P | R | F1 | P | R | F1 | P | R | F1 |
| Bathroom | – | – | – | 0.823 | 0.764 | 0.789 | 0.799 | 0.750 | 0.768 | 0.820 | 0.554 | 0.652 | 0.800 | 0.620 | 0.688 |
| Bedroom | 0.700 | 0.530 | 0.583 | – | – | – | 0.871 | 0.916 | 0.883 | 0.272 | 0.402 | 0.318 | 0.561 | 0.844 | 0.663 |
| Living Room | 0.642 | 0.444 | 0.514 | 0.753 | 0.818 | 0.779 | – | – | – | 0.817 | 0.458 | 0.576 | 0.817 | 0.458 | 0.576 |
| Kitchen | 0.887 | 0.365 | 0.506 | 0.991 | 0.569 | 0.715 | 0.950 | 0.444 | 0.593 | – | – | – | 0.943 | 0.467 | 0.614 |

perfect precision but random interaction renders very low recalls and F1 scores. Compared to the ablated versions, we validate the effectiveness of our learned binary-prior, scene-prior, the proposed fast-interactive-adaptation strategy, and the self-supervised learning-from-exploration policy. It also meets our expectation that the performance improves with more interactions (*i.e.*, from 10% to 20%), which is further validated by Fig. D.12. We show qualitative results in Fig. 4 and analyze them below.

**Is the Learned Prior Knowledge Useful?**  In Table 1, we see that **Ours-PriorOnly** already gives decent performance even without performing any interaction, outperforming **Random** by a large margin, which proves the usefulness of the learned prior knowledge. As shown in Fig. 4 (the left-most figure in the bottom row), the prior knowledge correctly predicts many relationships (red lines) to the ground-truth ones (blue lines). In addition, both BR-Prior-Net and SR-Prior-Net are helpful, as observed in the ablated versions **Abla-NoBinaryPrior** and **Abla-NoScenePrior**. We find that the BR-Prior-Net makes a lot of false-positive predictions and the SR-Prior-Net successfully removes them given the scene context. The performance drop of **Abla-NoBinaryPrior** further illustrates the need for binary priors, which initialize SR-Prior-Net with a sparse graph, better than a dense one.

**Does Fast-Interactive-Adaptation Help?**  The performance of **Ours-PriorOnly** is further improved given test-time interactions and posterior adaptions as **Ours-Final** generates better numbers. In Fig. 4, we see that some redundant red lines are removed observing the interaction outcomes and applying the SR-Posterior-Net predictions. Compared to **Abla-RandomAdapt** that performs random interactions, we find that the fast-interactive-adaptation strategy performs better. Qualitatively, besides the stove case mentioned in Fig. 1 (right), another interesting case in Fig. 4 is that from prior knowledge the agent thinks the two knives (ID: 57/67) may cut egg (ID: 50), maybe because it looks like a tomato (ID: 62) in geometry. The agent only interacts one knife (ID: 57) to correct this wrong belief and then the false-positive edge from the other knife (ID: 67) is automatically removed.

**Is Exploration Policy Effective?**  We learn an exploration policy for efficient exploration of the training scenes. **Abla-RandomExplore** gives a clear performance drop when using randomly collected data for training, proving that the exploration policy can wisely pick objects to explore.

## 5.4  TRANSFER TO NOVEL ROOM AND OBJECT TYPES

We further test our model's generalization capability to objects and rooms of novel types. In Table 2, we report quantitative evaluations where we train our systems on one type of AI2THOR rooms and test over the other types, as well as additional results of testing over a totally out-of-distribution RoboTHOR (Deitke et al., 2020) test dataset (see Fig. A.7). We find that 1) it works in most cases showing that our learned skills can transfer to novel room types; 2) our system can also generalize to novel object categories. For example, as shown in Fig. 5, although the agent never sees the television nor remote in the training scenes of bedrooms, it thinks the TV might be controlled by the remote. 3) the prior knowledge is not working well because of the data distribution shift but the posterior interaction helps. For example, in Fig. 5, the interaction helps to make accurate predictions despite the prior is not very accurate.

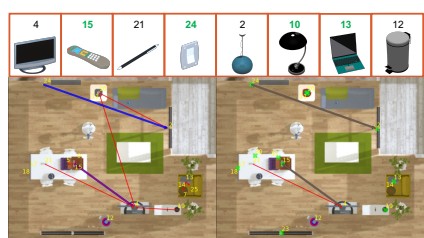

Figure 5: Bedrooms to living rooms.

## 6  CONCLUSION

We formulate a novel task of learning inter-object functional relationships in novel 3D indoor environments and propose a two-staged interactive learning-from-exploration approach to tackle the problem. Using a newly created hybrid dataset, we provide extensive analysis of our method and report quantitative comparisons to baselines. Experiments comparing to several ablated versions validate the usefulness of each proposed module. See appendix for limitations and future works.

**Ethics Statement.** We explore AI algorithms towards building future home-assistant robots which can aid people with disabilities, senior people, etc. While we envision that our proposed approach may suffer from the bias in training data, the privacy of using human-designed data, and domain gaps between training on synthetic data and testing in real-world environments, these are the common problems that most if not all learning-based systems have. We do not see other particular harm or issue of major concerns that our work may raise.

**Reproducibility Statement.** We support open-sourced research and will make sure that our results are reproducible. Thus, we promise to release the code, data, and pre-trained models publicly to the community upon paper acceptance.

**Acknowledgements** This paper is supported by a Vannevar Bush Faculty fellowship and NSF grant IIS-1763268.

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

APPENDIX

## A    MORE DATASET DETAILS

Fig. A.6 shows one example room from each of the four types in AI2THOR (Kolve et al., 2017). Each AI2THOR scene is designed manually by 3D artists from reference photos. There is a randomizer that can be used to change the location of the objects, so as to enrich the scene. RoboTHOR (Deitke et al., 2020) contains 25 test scenes that have different styles from the AI2THOR scenes (see Fig. A.6). Fig. A.7 shows some RoboThor scenes we are using. We can see that the scenes are visually different and thus we consider them to be two different distributions of scenes.

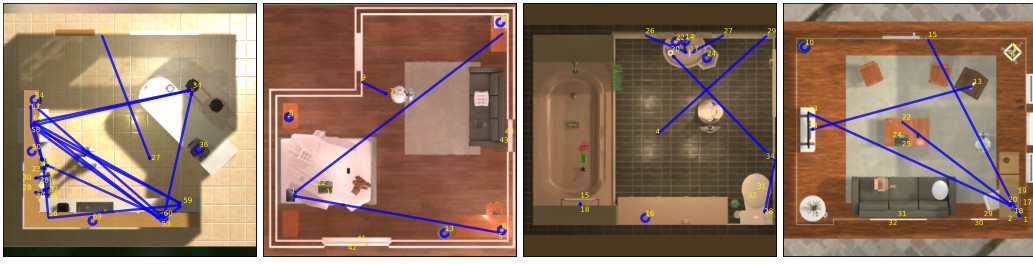

Figure A.6: Four room types of AI2Thor scenes. From left to right: kitchen, bedroom, bathroom, living room. The yellow numbers are object index, the blue lines are IFR.

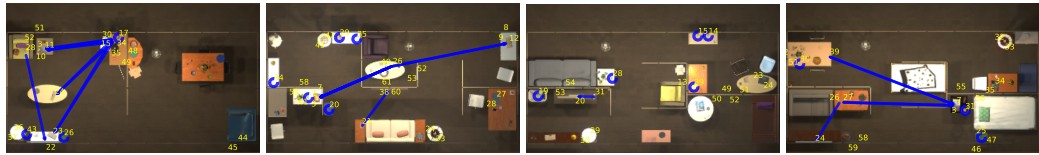

Figure A.7: RoboThor data gallery.

Table A.3 presents the detailed data statistics. We collect all AI2Thor models and scenes offline and build our own simulation environment with standard OpenAI Gym API (Brockman et al., 2016). We construct the edges of a scene by grammar based on object classes. For many-to-many relationships

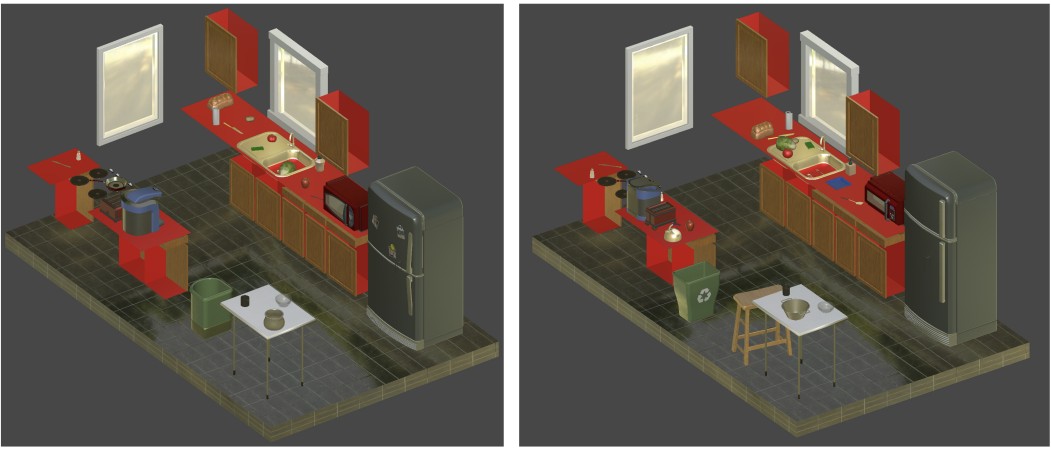

Figure A.8: Two FloorPlan 11 variances in our dataset.

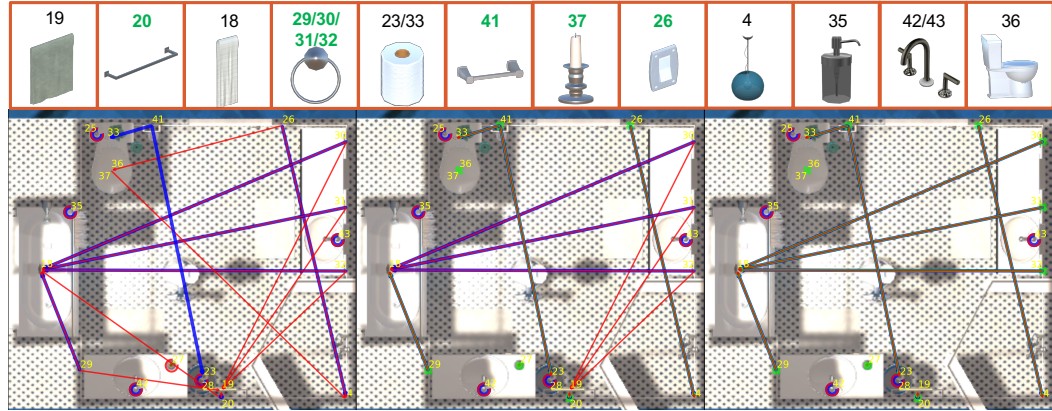

Figure B.9: A qualitative result in the bathroom. There are many holders used for different purposes. For example, the bath towel holder (ID: 20) is used for bath towel (ID: 19); the hand towel holder is (ID: 29/30/31/32) used for the hand towel (ID: 18); the toilet paper holder is (ID: 41) used for toilet paper (ID: 23/33). Those holders look similar and the agent fails to give accurate predictions based on prior knowledge. However, the agent knows those holders are hard to distinguish and use the fast-interactive-adaptation policy to reduce the uncertainty.

(*e.g.*, knives can cut potatoes), all triggers are connected to responders. For one-to-one relationships (*e.g.*, the buttons open lamps), we follow the rules of AI2Thor if possible. For cases that AI2Thor suffices (*e.g.*, there is no ceil lamp instance in the AI2Thor scenes), we manually bind the relationships with heuristic rules based on distances and human knowledge. Among all the 48 edge types, 19 are one-to-one and 29 are many-to-many. If the object should have detailed parts (*e.g.*, Microwave), we substitute the AI2Thor mesh with the PartNet model.

Table A.3: data statistics

|  |  | #scenes | #shape-cats | #objects | #int-objects | #relations |
|---|---|---|---|---|---|---|
| AI2THOR + PartNet | Kitchen | 300 | 63 | 20880 | 4060 | 6790 |
|  | Bathroom | 300 | 40 | 10850 | 3540 | 3620 |
|  | Bedroom | 300 | 59 | 12280 | 2240 | 2520 |
|  | Living Room | 300 | 50 | 12710 | 1590 | 1830 |
| RoboThor + PartNet |  | 60 | 49 | 3104 | 488 | 579 |

# B    MORE QUALITATIVE RESULTS

We show more qualitative results in this section. In Fig. B.9 we show an example inference in the bathroom. There are many holders used for different purposes. For example, the bath towel holder (ID: 20) is used for bath towel (ID: 19); the hand towel holder is (ID: 29/30/31/32) used for a hand towel (ID: 18); the toilet paper holder is (ID: 41) used for toilet paper (ID: 23/33). Those holders look similar and the agent fails to give accurate predictions based on prior knowledge. However, the agent knows those holders are hard to distinguish and uses the fast-interactive-adaptation policy to reduce the uncertainty. In Fig. B.10, we show an example inference in the bedroom. We can see that the prior knowledge is already good without any interaction. And with interaction, the performance can be further improved. For example, the agent learns to turn on the ceiling lamp (ID: 3) by toggling a novel button (ID: 9) that doesn't appear in the training scenes.

# C    DIFFERENT RELATIONSHIP ABLATION.

In this section we discuss the agent's performance over difference types of relationships. As shown in Table C.4. Overall the agent performs better on the Many-to-Many relationship, which is reasonable because One-to-One relationships would suffer from ambiguities. Another clear trend is that the agent performs much better in the Self relationship compared to the Binary relationship, which indicates the difficulties to build IFRs across objects.

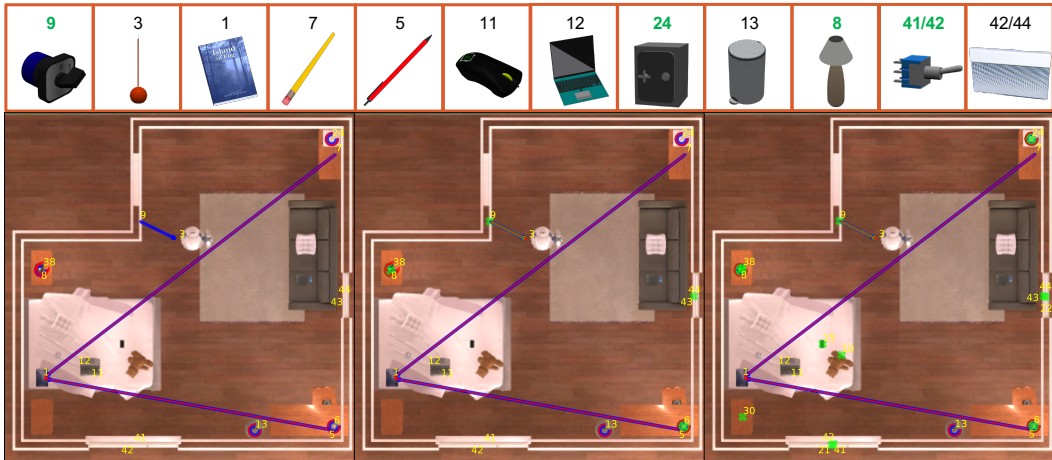

Figure B.10: A qualitative result in the bedroom. We can see that the prior knowledge is already good without any interaction. And with interaction, the performance can be further improved. For example, the agent learns to turn on the ceiling lamp (ID: 3) by toggling a novel button (ID: 9) that doesn't appear in the training scenes. We also show the overall performance for reference.

Table C.4: We present performance over different types of relationships. One-to-One and Many-to-Many are classified by number of links of objects of GT graph. Self means to trigger one object leads to state change of itself. Binary means to interact one object leads to state change of another object.

| Method | Bathroom | | | | Bedroom | | | | Living Room | | | | Kitchen | | | |
|---|---|---|---|---|---|---|---|---|---|---|---|---|---|---|---|---|
| | P | R | F1 | MCC | P | R | F1 | MCC | P | R | F1 | MCC | P | R | F1 | MCC |
| Overall | 0.937 | 0.984 | 0.957 | 0.958 | 0.877 | 0.987 | 0.924 | 0.927 | 0.745 | 0.965 | 0.833 | 0.843 | 0.848 | 0.890 | 0.864 | 0.865 |
| One-to-One | 0.935 | 0.983 | 0.955 | 0.957 | 0.935 | 0.983 | 0.954 | 0.956 | 0.741 | 0.966 | 0.831 | 0.841 | 0.857 | 0.812 | 0.819 | 0.826 |
| Many-to-Many | 0.968 | 0.992 | 0.977 | 0.978 | 0.758 | 1.000 | 0.847 | 0.862 | 1.000 | 0.900 | 0.933 | 0.941 | 0.867 | 0.934 | 0.893 | 0.896 |
| Self | 0.993 | 0.992 | 0.992 | 0.991 | 0.927 | 1.000 | 0.956 | 0.957 | 0.832 | 0.985 | 0.879 | 0.889 | 0.998 | 0.830 | 0.898 | 0.900 |
| Binary | 0.908 | 0.978 | 0.935 | 0.939 | 0.867 | 0.982 | 0.915 | 0.919 | 0.736 | 0.966 | 0.826 | 0.838 | 0.823 | 0.905 | 0.856 | 0.859 |

## D  FAST-INTERACTIVE-ADAPTATION BEHAVIOR

Here we show the behavior of fast-interactive-adaptation policy. In Fig. D.11, we visualize the distribution of objects that the fast-interactive-adaptation policy interacts with. We can see that the adaptation policy tends to interact with interesting objects with IFR (*e.g.*, switches and knobs). In Fig. D.12, we can also see the performance steadily improve with the number of interactions.

## E  INITIALIZATION OF EDGES IN GCN.

Here we discussed the edge initialization of GCN of SR-Prior-Net and Exploration Policy. Table E.5 shows that: 1) the performance drops with only prior knowledge or distance information as edges; 2) a scaling factor can boost the performance. Our hypothesis is that the scaling of initial edges helps the model to distinguish them from GT edges, thus making it easier to do casual inference.

## F  FAILURE CASES AND ANALYSIS

There are cases where the object shapes are very similar, but the IFR is different. For example, Fig. F.13 shows one failure case in a bathroom scene, where the agent confuses the hand towel with the bath towel and predicts two extra edges. (the hand towel holder is used to hold the bath towel and the bath towel holder is used to hold hand towel.)

## G  LIMITATIONS AND FUTURE WORKS.

First, our work uses 3D point clouds as inputs, ignoring the object color, texture, or other non-geometric information. Besides, our system assumes all the objects in the scene are pre-segmented and many visual quantities are direction observable (*e.g.*, the state change of objects). Future

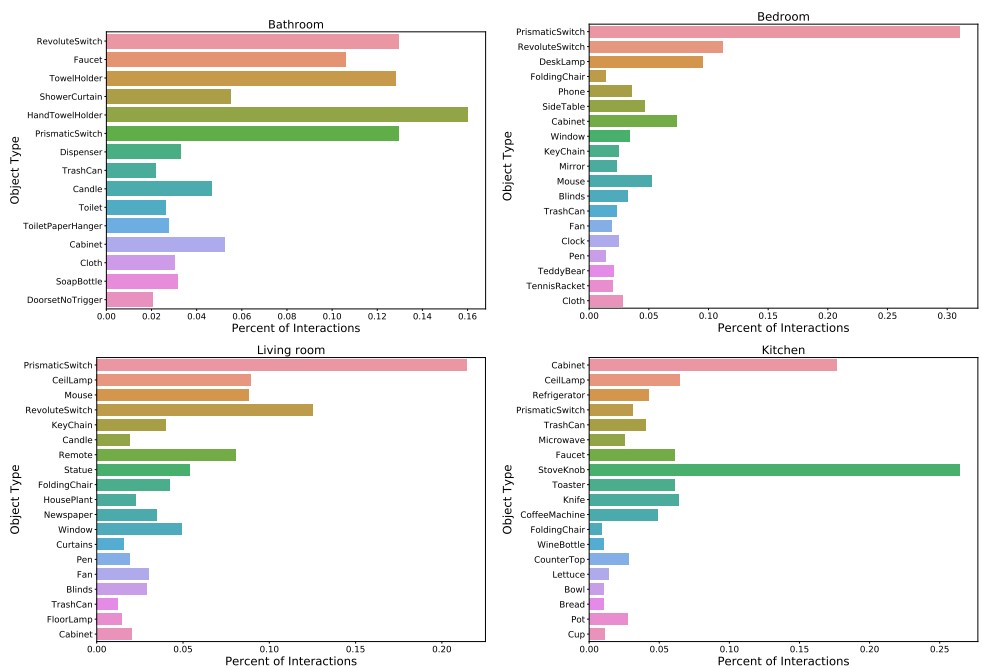

Figure D.11: The distribution of objects that the fast-interactive-adaption policy interacts with on the test scenes. We discard objects with less than 10 interactions in the whole test scene.

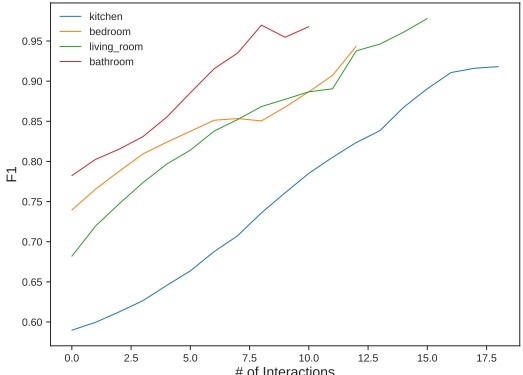

Figure D.12: The average F1 score in the test scenes with respect to the number of interactions.

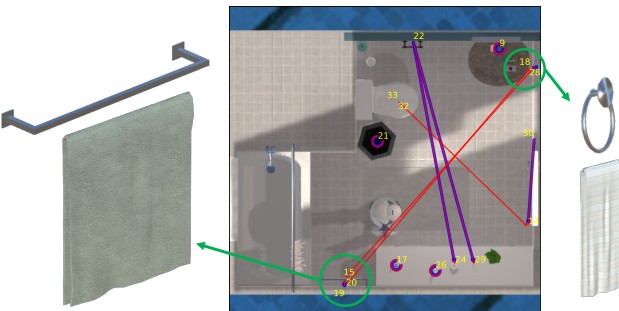

Figure F.13: A bathroom scene where a hand towel holder holds a hand towel while a bath towel holder holds a bath towel. The agent confuses the two towels and predicts two extra edges.

Table E.5: The edge initialization of GCN of SR-Prior-Net and Exploration Policy. The experiments is conducted on the bathroom with 20% budget.

|  | Precision | Recall | F1 |
|---|---|---|---|
| Prior only | 0.872 | 0.971 | 0.913 |
| Distance only | 0.835 | 0.986 | 0.902 |
| No Scaling ($\gamma = 1.$) | 0.819 | 0.991 | 0.894 |
| Scaling factor $\gamma = 0.4$ | 1.0 | 0.836 | 0.908 |
| Scaling factor $\gamma = 0.8$ | 0.873 | 0.986 | 0.922 |
| Ours-Final | 0.873 | 0.989 | 0.923 |

works can study how imperfect visual observations may affect our network predictions and how to address such challenges if any. Lastly, we also abstract away the robotic navigation and detailed manipulation. Future works may investigate building an end-to-end pipeline considering navigation and manipulation.

