# OpenReview forum: "IFR-Explore: Learning Inter-object Functional Relationships in 3D Indoor Scenes"
_ICLR.cc/2022/Conference — ICLR 2022 Poster_

### Official Review · Reviewer_TQTu · 2021-10-18

**Correctness:** 4
**Technical Novelty And Significance:** 3
**Empirical Novelty And Significance:** 3
**Recommendation:** 8
**Confidence:** 4

**Details Of Ethics Concerns:**

The authors address these concerns properly in the paper. No ethics concern at the reviewer's end.

**Main Review:**

These are some solid points in the paper:
1. The paper introduces the extraction of the inter-object functional relationship as a novel problem to explore.

2. There is a new hybrid dataset composed of 1200 scenes with 23,360 objects from 121 categories to study this problem.

3. The paper introduces a strategy based on PointNet++ to infer the binary inter-object functional relationship, a graph convolutional network to model the scene and predict the functional graph for the scene.

4. A significant effort has been spent to showcase different aspects of the solution.

5. The authors promise to make their code and dataset available upon acceptance.

These are suggestions I make to improve the paper:

1. The authors should add results to their abstract.

2. The statement  “Performs as few interactions as possible to reduce uncertainties” should be quantified.

3. Please define PPO (proximal policy optimization?) in the text.

4. In the following statement: “In our implementation, we train the exploration policy and the prior/posterior networks for five loops until they jointly converge.” Please clarify: it is five loops or until convergency?

5. I did not understand this sentence: “on average 32% of the total object count 3125 in all 210 training scenes.” Please clarify.

6. I have several comments on the sentence: “We replace 83 AI2THOR models with 888 PartNet-Mobility models for 31 categories (12 exist in AI2THOR while 19 are newly added) to further enrich the diversity of the shape geometry.”

a. I suggest replacing “models” with “objects” for clarity.

b. Would you please elaborate on the need to modify the AI2THOR dataset? Would you please elaborate on the diversity of shape geometry you are aiming for?
c. I am intrigued by how do you go from 120 interactive scenes to 1200? What are the criteria to create a new scene? How does one scene differ from the rest? Would you please quantify the difference between scenes?

7. I suggest you bold the best results in Table 1.

**Summary Of The Paper:**

The paper proposes a framework to study inter-object functional relationships, i.e., where the state change in one of the objects causes functional effects on other 3D objects.  The authors propose a two-stage strategy guided by a self-supervised strategy. The first stage predicts possible functional relationships between the objects. In the second stage, the system performs interactions between the objects in the scene. The authors present the inter-functional relationship with a directed graph.

The binary relationships are represented with a prior network. Processing all the objects results in a scene representation of a functional relationship. Their procedure for training an exploratory policy allows them to generate data to construct a posterior network of the functional relationship in the scene. The authors create a dataset using scenes and objects from AI2Thor, enriched with articulated objects from PartNet.

The authors present an ablation study where baselines are extracted for the binary priors, scene priors, the exploration policy, the selection of objects, and their level of interaction.

**Summary Of The Review:**

The paper introduces a novel and relevant problem. In the process, the authors construct a dataset for experimentation based largely on previous ones. The paper presents a strategy to solve the problem while the experimentation showcases different aspects of the solution. Finally, the authors promise to make their code and dataset available upon acceptance. Still, some issues need to be addressed to clarify the message in the paper further.

---

> ### Author Response · Authors · 2021-11-22
> **Response to Reviewer TQTu: Thank you so much for the positive feedback and valuable suggestions! (Part 1/2)**
>
> Thank you so much for the positive feedback and valuable suggestions for further improving our work! Thank you for taking the time and effort to review our paper!
>
> It is our great pleasure to hear that the reviewer recognizes our contributions to the novel problem we propose to address, the new hybrid dataset to study the problem, the technical method designs, and our results to showcase different aspects of the solution. We will release our code and data for the reproducibility of the work and hope to draw more research attention to this underexplored yet important task in embodied AI.
>
> **Thank you for your valuable suggestions to improve the paper! We have addressed all of them below and revised the paper accordingly to reflect the changes (we highlight all changes in red in the revised paper)!**
>
> > Add results to the abstract.
>
> Thank you for the suggestion! We have added two sentences of our result summary in the abstract of the revised paper: “Results show that our model successfully learns priors and fast-interactive-adaptation strategies for exploring inter-object functional relationships in complex 3D scenes. Several ablation studies further validate the usefulness of each proposed module.”
>
> > Quantity the statement “performs as few interactions as possible to reduce uncertainties”
>
> Thank you for the suggestion! In our experiments, for fair comparisons among different methods, we set a fixed test-time interaction budget as 10% or 20% of the total number of objects in each room, as we stated in the last sentence of Sec 4.2.2 and the caption of Table 1. If no preset budget is given, our method can automatically determine when to stop using Equation (3). When $\gamma=0.05$, we empirically find that the agent explores about 19% of objects in the room on average (we have revised the paper to add this). If one thinks this is too much, we can increase $\gamma$ to adjust.
>
> > Define PPO (proximal policy optimization?) in the text.
>
> Thank you for pointing this out! Yes, we use the proximal policy optimization algorithm. We have made this clear in the revised paper.
>
> > Clarify “it is five loops or until convergence?”
>
> This is a great point. Sorry, we didn’t make this clear and have revised the text. We aim to train the networks jointly until they converge. We empirically find that five loops of training are enough for the exploration policy and prior/posterior networks to converge jointly.
>
> > Clarify the sentence “on average 32% of the total object count 3125 in all 210 training scenes”.
>
> Thank you for pointing this out. We are sorry that we didn’t make it clear and have revised the text to “e.g., roughly 32% of the total 3125 objects from 210 scenes”. We intended to give the readers a rough idea of what percentage of objects in the rooms have been explored given the 1000 total interaction budget. The interaction budget 1000 takes up 32% of the 3125 objects in 210 training scenes that the agent visited (i.e., 1000/3125=0.32). We report this also to show that the interaction budget 1000 only allows exploring a subset of objects, not all of them, so our exploration strategy needs to be intelligent and efficient.

---

> > ### Author Response · Authors · 2021-11-22
> > **Response to Reviewer TQTu (Part 2/2)**
> >
> > > On the sentence: “We replace 83 AI2THOR models with 888 PartNet-Mobility models for 31 categories (12 exist in AI2THOR while 19 are newly added) to further enrich the diversity of the shape geometry.” a) replace “models” with “objects”; b) elaborate the need to modify the AI2THOR dataset and the diversity of shape geometry; c) how to go from 120 to 1200?
> >
> > Thank you for the detailed and useful suggestions!
> >
> > For a), we agree with the reviewer and have revised the paper to use the word “objects” instead of “models”. Thank you!
> >
> > For b), the primary reason to enrich the AI2THOR scenes with more PartNet-Mobility objects is to build a large-scale dataset containing diverse 3D object geometry and categories for training our system over large-scale, diverse 3D objects and thus better generalizable to novel object shapes and categories in test scenes. The original AI2THOR scenes are great for our needs but the total number of distinct objects in the scenes that are of interest to our problem is quite few and not enough. For example, there are only 5 different faucets in the original AI2THOR kitchens but 84 in PartNet-Mobility; and, there are only 8 different spray bottles in AI2THOR but 57 in PartNet-Mobility; etc. Also, many interesting categories (e.g., mouses, lighters, scissors, etc) are lacking. Our system takes the raw shape geometry as inputs and aims to generalize over large-scale, diverse object shapes. For example, there are so many different kinds of faucets in the real world. Only training over 5 AI2THOR faucets cannot learn a perception system that generalizes to various different instances of faucets with different geometry. Therefore, we replace 83 AI2THOR objects with 888 PartNet-Mobility objects to enlarge the diversity of object shapes for training a better model of our framework. We also introduce 19 new object categories to enrich the dataset since we also aim to train the system to be generalizable to novel unknown object categories at the test time. We are happy to add these elaborations in the paper revision if requested and would like to clarify more if needed.
> >
> > For c), we used two ways to create new scenes: 1) since each of the 120 AI2THOR scenes is associated with a randomizer that can be called to change the location of the objects, we can get multiple different scene instances for each AI2THOR scene by calling the randomizer for several times, and 2) we can replace the object geometry of one object with another one of the same types from the PartNet-Mobility dataset (e.g., a different microwave). **We have revised the paper to make this clearer** (see the second paragraph of Sec. 5.1). **See Figure A.8 for examples of different scene variations we created based on the same AI2THOR scene.** It is a bit unclear to us how to quantify the scene differences. However, we are happy to provide quantitative numbers if instructed.
> >
> > > Bold the best results in Table 1.
> >
> > Thank you for pointing this out! We have modified Table 1 in the revised paper.
> >
> > **We sincerely hope that our response above makes things clearer to you and addresses your suggestions well. Otherwise, please do not hesitate to ask us more and we are very happy to discuss further. Thank you again for the useful comments and questions!**

---

> ### Comment · Reviewer_TQTu · 2021-11-29
> **overall opinion on the paper**
>
> I have read the answers to my remarks, read the changes made to the paper, and followed the discussion carried out with other reviewers. Overall, I am still satisfied with the authors' article and sustain my opinion.
>
> Thanks

---

> > ### Author Response · Authors · 2021-11-30
> > **Thank you!**
> >
> > Thanks for your final positive rating. We are glad that our responses help alleviate your concerns. Thanks again for all your valuable feedback!

---

### Official Review · Reviewer_sxhc · 2021-11-02

**Correctness:** 3
**Technical Novelty And Significance:** 2
**Empirical Novelty And Significance:** 2
**Recommendation:** 6
**Confidence:** 4

**Main Review:**

===Quality of Results===:

S: Extensive experiments on the design choices have been performed, which informs the reader of the need and functioning of incorporating such modules. The proposed method is a modular approach with simple networks, which is quite easy to follow.

W: Well, it is difficult to objectively judge how good the results are. Although P, R and F1 scores have been used to measure the quality of results, they do not necessarily capture the "goodness" score, i.e,  P and R scores are biased, or rather, hide important flaws in the data stats that have a significant bearing on the result interpretation (and therefore the Model's learned ability). How about Matthew's Correlation Coefficient (MCC)? This metric correctly captures the quality of results and sidesteps issues prevalent with P and R scores.



===Comparisons===:

S & W: There is no comparison to prior works since the claim is that this is the first paper proposing this problem.
However, experiments have been performed to validate the system design choices, such as the need for individual prior nets (binary, as well as, scene).


===Novelty===:

W: Problem statement -- The claim is that the problem is new. Application-wise, it may be, but it is nothing more than the task of link prediction. As such, I am not sold on the novelty of the problem.

Neutral: Approach -- Good modular approach, but expected.

Neutral: Results -- Interpretation could be better.



===Limitations===:

Problem novelty and Evaluations, as described above.





===Relevant Literature Coverage===:

S: "Related work" section is comprehensive and covers a lot of relevant papers in this field.



===Writing===:

S: The writing is good.



**Summary Of The Paper:**

The paper presents an interesting research problem of "Learning Inter-Object Functional Relationships" in virtual 3D indoor environments, where the goal is to make an agent understand the functional effect on an object by interacting with other objects in the scene. Two important (and related) questions arise: (a) How to understand which object(s) affects the functionality of which other object(s) in the room?, and (b) how to learn such inter-object functional priors to build a good knowledge base for reliable inference in unseen environments?

The paper addresses these two questions by presenting a reinforcement-based learning algorithm.

The technical summary of the paper is given below.

===Goal===:

Learning inter-object functional relationships, represented by scene graphs. That is, an agent is provided with large-scale scenes to explore for learning in the training stage and is asked to predict a functional scene graph, (S, R_s) for a new scene at test time. Here S is the scene and R_s is the relationship scene graph for the scene, S.



===Assumptions===:

Object segmentation and state changes (by "state", the authors refer to the functional outcome of an object) are provided to the agent apriori. Therefore, the functional effect of a relationship is directly observable.



===Input===:

A 3D scene as a set of objects {O_1, O_2, ..., O_n}, where each object O_i is explicitly modeled as (O_i^hat, c_i, s_i) where O_i^hat represents the point cloud of the object, c_i represents the 3D centroid, and s_i represents the object's isotropic scale.



===Output===:

Functional scene graph, R_s, i.e., whether there exists a link between two nodes (objects) of the scene graph (scene).




===Learning Style===:

Self-supervised learning framework




===Neural Network===:

1) Pretrained PointNet++ and an MLP (these two make up what is called a "BR-Prior-Net", which is short for Binary Relationship Prior Net) -- supervised training using observed GT data

2) GCN, 3-layer (this is a part of what is called an "SR-Prior-Net", which is short for Scene Relationship Prior Net) -- Partly supervised using data from the above step and some rule-based heuristics

The goal of the above two networks is to propose functionally related objects.

3) SR-Posterior-Net, which has the same architecture and weights as SR-Prior-Net. The only difference is that the input to SR-Posterior-Net, in the beginning, is the refined prediction output from SR-Prior-Net.

4) A GCN trained using Reinforcement Learning (RL) algorithm to output action probability scores for each object that indicates the probability that this object triggers a state change on another object. Essentially, the RL-algorithm-based optimization on GCN is used to obtain supervision ground truth.



===Dataset Used===:

AI2Thor, PartNet-Mobility

1) A total of 1200 scenes covering 23360 objects from 121 categories are used from the above datasets.

2) Among these, 27 are trigger objects, 29 are responders, where 11 are both trigger and responders (e.g., desk lamp), and 79 are non-interactive background objects



===Relevance of datasets===:

The above two datasets employed fit well for working on the proposed problem.


===Loss===:

1) For training the RL algorithm, a Policy Optimization loss is formulated.
2) SR-Priori, BR-Priori, and SR-Posteriori are all simple link prediction tasks. So, standard binary cross entropy loss.



===Objective Evaluation Metrics===:

Precision, Recall and F1-scores

**Summary Of The Review:**

Essentially, the problem of inferring inter-object functionality boils down to link prediction tasks, but the meat lies in the fact that supervisory GT signals are mined from the outputs of a developed system within.

I give credit to the self-supervision part, but then, the problem itself pivots towards the theme of extracting priors from data for learning a different task. Plus, the evaluation is not reliably interpretable with the results shown. As such, I am sitting on a borderline score for this paper.

---

> ### Author Response · Authors · 2021-11-22
> **Response to Reviewer sxhc: Thank you so much for the positive feedback and valuable suggestions! (Part 1/2)**
>
> Thank you so much for the positive feedback and valuable suggestions for further improving our work! Thank you for taking the time and effort to review our paper!
>
> It is our great pleasure to hear that the reviewer recognizes our extensive experiments over the design choices and our self-supervised learning framework for a new and interesting problem. We are also very happy to hear that the reviewer thinks our modular approach is easy to follow, the related works discussed are comprehensive, and the writing is good. We will release our code and data for the reproducibility of the work and hope to draw more research attention to this underexplored yet important task in embodied AI.
>
> **Thank you for your valuable suggestions to improve the paper! Thank you for the questions! We have addressed all of them below and revised the paper accordingly to reflect the changes (we highlight all changes in red in the revised paper)!**
>
> > Precision and recall may be biased and hide important flaws. Report result evaluations using the “Matthew's Correlation Coefficient (MCC)”?
>
> Thank you so much for this very nice suggestion and we fully agree with the reviewer! **In the revised Table 1 of the paper, we report the MCC evaluations and comparisons.** We observe that our method still performs the best. We hope that adding the MCC results helps better interpret our results. We are happy to take more advice from the reviewer on improving the evaluations and results interpretations if there are any.
>
> > No comparison to prior works since the claim is that this is the first paper proposing this problem.
>
> To our best knowledge, we do not know the other works that address the same problem and can be used for direct and fair comparisons. However, **we try our best to adapt Nagarajan & Grauman (2020) to our setting**, though it originally tackles a very different task on learning affordance over individual shapes in a scene and does not propose IFRs. We use a PointNet++ to encode object features and an MLP to additionally predict a set of objects that each interaction at a pixel causes. **We add this new baseline to Sec. 5.2 and report the quantitative comparisons in Table 1 of the revised paper**, from which we see that **our method still performs the best**. We would be very happy to add other comparisons if the reviewer may suggest any.
>
> Thank you for recognizing our ablation studies for validating the system design choices! We hope that these ablated methods also serve well as the benchmarking baselines for comparisons.
>
> > Though a new application and using self-supervised learning, the problem belongs to the link prediction problems? The novelty of the problem?
>
> Admittedly, if we treat the objects as nodes and the inter-object functional relationships as edges, our problem is indeed a special case of the general type of link prediction problems. However, **the specific problem we are studying is an important yet underexplored task in recent embodied AI research**, which motivates us to propose this novel problem and raise up people’s research attention on this, and **there are also many non-trivial and problem-specific technical designs to tackle this challenging task**, on which we would like to claim our contributions.
>
> First of all, as the reviewer already noticed, how to effectively and efficiently explore the training scenes given limited interaction budgets for self-supervision during training is a big challenge and the proposed interactive exploration RL policy that is carefully customized for the task is one of our main contributions. We think that the capability of collecting data via interactions for self-supervised learning is necessary and crucial for future embodied AI autonomous agents.
>
> Secondly, it is also quite challenging to figure out a working solution over large-scale 3D objects and scenes with diverse geometry and rich semantics, aiming for generalization to novel scenes. Many careful problem-specific designs are involved and non-trivial, such as training over large-scale 3D geometry for generalization, decoupling a binary-relationship network from the scene-level one to reduce the problem complexity, allowing test-time interactions to reduce intrinsic uncertainties in novel test scenes, and consequently requiring a posterior network that can adapt the predictions given new test-time interaction observations.

---

> > ### Author Response · Authors · 2021-11-22
> > **Response to Reviewer sxhc (Part 2/2)**
> >
> > Last but not least, for future home-assistant robots to tackle embodied tasks like our problem, we would like to argue that it is very important to combine the learned priors over large-scale training data and a test-time fast-interactive-adaptation strategy for posteriors to adapt to novel scenes. The large-scaled prior learning stage can learn the most common-sense knowledge about 3D objects and their functionalities in daily human environments, while the test-time adaptation stage is also crucial for fast probing the new environment, addressing possible intrinsic uncertainties, and producing good results for novel scenes.
> >
> > **We sincerely hope that our response above makes things clearer to you and addresses your suggestions well. Otherwise, please do not hesitate to ask us more and we are very happy to discuss further. Thank you again for the useful comments and questions!**

---

### Official Review · Reviewer_RFiX · 2021-11-09

**Correctness:** 4
**Technical Novelty And Significance:** 3
**Empirical Novelty And Significance:** 3
**Recommendation:** 6
**Confidence:** 3

**Main Review:**

Strengths:
-The task is interesting. Although I do not think exploring inter-object functional relationships in 3D indoor scenes is a new task (some previous works about causality modeling proposed similar tasks, such as "Learning Perceptual Causality from Video"), I appreciate the collected dataset and the designed pipeline of learning the relationships for inter-object.

-The dataset and the proposed pipeline may inspire some applications such as home-assistant robots, intention prediction, and so on.

Weaknesses:
-Some details are missed which may affect the reproducibility o the work. For example. the reward design and shaping are not discussed. The FAST-INTERACTIVE-ADAPTATION is a little confusing. I am not sure what "fast" refers to or what makes the framework fast. It is nice if the authors could clarify this point.

-The motivation of the design of some components could be discussed more clearly. For example, it is not intuitive for me why the input of binary-relationship prior network is a point cloud. It is not clear for me either why two objects nodes are connected because they are near(<0.5m)

-The experiment does not evaluate the proposed framework over different types of relations. For example, I am wondering what is the performance of the framework on the many-to-many relationship prediction.




**Summary Of The Paper:**

This paper proposes to study inter-object functional relationships such as a switch on the wall turning on or off the light. To tackle this task, this work designs a two-stage approach under a self-supervised interactive learning-from-exploration framework, which can predict functional relationships between objects in new 3D indoor environments. Furthermore, this work creates a dataset based on AI2THOR and AI2THOR to demonstrate the task.

**Summary Of The Review:**

In summary, the task of this work is interesting and I think it gives some insight into exploring the general functionality of objects. The paper is well organized. I would be inclined to suggest accepting it if the motivation and some technical details could be clarified in the revision.

---

> ### Author Response · Authors · 2021-11-22
> **Response to Reviewer RFiX: Thank you so much for the positive feedback and valuable suggestions! (Part 1/2)**
>
> Thank you so much for the positive feedback and valuable suggestions for further improving our work! Thank you for taking the time and effort to review our paper!
>
> We are very happy to hear that the reviewer thinks our proposed task is interesting and the paper is well organized. We will release our code and data for the reproducibility of the work and hope to draw more research attention to this underexplored yet important task in embodied AI.
>
> **Thank you for the valuable comments and suggestions! Thank you for pointing out the missing details and motivations of our paper! We believe that adding these would be very important, giving more insights to the readers. We have addressed all of them below and revised the paper accordingly to reflect the changes (we highlight all changes in red in the revised paper)!**
>
> > Discuss more details about the reward design and shaping.
>
> Thank you for pointing this out! **We have added more details about the reward design and shaping in the revised paper** (see the highlighted text in red in the second paragraph of Sec. 4.3.1). We hope that these additional details regarding the reward design and shaping make things clearer to the reviewer.
>
> For your quick reference, we use the reward function defined as follows.
>
> $$
> reward^{(t)} = \alpha \max_j |r_{i,j}^{(t)}-e_{i,j}|  + \beta \mathbf{1}[e_{i,j}] - \gamma
> $$
>
> The first term encourages the agent interacting with an object $O_i$ to correct the wrong or uncertain functional relationship beliefs $r^t_{i,j}, \forall j$. There are three cases for this term:
>   -  the agent's belief is already correct ($r^t_{i,j} = e_{i,j}$, thus $|r^t_{i,j}-e_{i,j}|=0$);
>   -  the belief is wrong ($r^t_{i,j} = 1 - e_{i,j}$, thus $|r^t_{i,j}-e_{i,j}|=1$);
>   -  the agent is uncertain (e.g., $r^t_{i,j} = 0.5$, thus $|r^t_{i,j}-e_{i,j}|=0.5$).
>
> We use _max_ instead of _avg_ as the aggregation function because the IFRs are sparse and the average value may conceal the mistake.
>
> The second term rewards exploring trigger objects of interest, e.g., it encourages the agent to interact with buttons or switches, instead of boxes or cabinets, which may be thought of as a reward shaping term.
>
> The third term simulates the interaction cost to avoid redundant interactions for efficient exploration. For the tunable hyperparameters, we use $\alpha=2$, $\beta=1$ and $\gamma=1$ in our experiments.
>
> > What does “fast” mean in the term “fast-interactive-adaptation”?
>
> We are sorry for the confusion. Given a novel scene at the test time, our network first uses the learned prior knowledge to get initial prior predictions of the inter-object functional relationships, and then our agent may need few-shot test-time interactions in the scene to address some intrinsic uncertainties (e.g., if there are two buttons and lights in the room, which button turns on which light?) for more accurate posterior predictions. We employ a fast-interactive-adaptation strategy to do this and the word “fast” refers to that we only need to fastly do a very few interactions on objects with ambiguities to address, instead of having to explore the scenes for a long while interacting with many objects in the scene.
>
> > Why the input of binary-relationship prior network is a point cloud?
>
> The input of the binary-relationship prior network is a pair of two object point clouds in the scene. The intuition behind this is that given two objects there is already much semantic or geometric prior information indicating their functional relationships even without considering the scene context. For example, we know that buttons trigger electrical devices in general. And, by only taking two objects as inputs, we reduce the problem complexity of inferring relationships among all objects in the scene to consider each pair of objects. Experiments show that first inferring the binary priors and then refining them given the scene context via the scene-relationship prior network is a good strategy that works well (see the ablation study **Abla-NoBinaryPrior** in Table 1).
>
> > Why two objects nodes are connected because they are near (<0.5m)?
>
> Thank you for the great question and we have revised the paper to make this clearer! We connect nearby objects because intuitively closer objects are more likely to have functional relationships, e.g., a stove knob is close to the corresponding stove burner. We use 0.5m because we figured out that this distance covers a large proportion of ground-truth relationships in our dataset. Specifically, 53.1% of the functionally related object pairs are between two objects closer than 0.5m. An ablation study presented in Table E.5 (“prior only” ablation v.s. “ours-final”) empirically shows that connecting nearby objects indeed helps.

---

> > ### Author Response · Authors · 2021-11-22
> > **Response to Reviewer RFiX (Part 2/2)**
> >
> > > Evaluate over different types of relationships?
> >
> > This is a great point and is also mentioned by reviewer fQs8! **We have added this and reported the performance over different types of relations** (i.e., “one-to-one”, “many-to-many”, etc.) in the revised paper (see Appendix Sec. C, Table C.4).
> >
> > The results show that the agent actually performs very well on the “many-to-many” relationship, which is reasonable because “one-to-one” relationships would suffer more from the uncertainties and ambiguities.
> >
> > **We sincerely hope that our response above makes things clearer to you and addresses your suggestions well. Otherwise, please do not hesitate to ask us more and we are very happy to discuss further. Thank you again for the useful comments and questions!**

---

> > > ### Comment · Reviewer_RFiX · 2021-11-30
> > > **Final comments**
> > >
> > > Thanks for the response to my questions. I appreciate the clarification and the corresponding revision.
> > > I have read other reviewers' comments and the authors' responses as well.
> > > I will keep my ratings.

---

> > > > ### Author Response · Authors · 2021-11-30
> > > > **Thank you!**
> > > >
> > > > Thanks for your final positive rating. We are glad that our responses help alleviate your concerns. Thanks again for all your valuable feedback!

---

### Official Review · Reviewer_fQs8 · 2021-11-09

**Correctness:** 4
**Technical Novelty And Significance:** 2
**Empirical Novelty And Significance:** 2
**Recommendation:** 6
**Confidence:** 3

**Main Review:**

The problem is interesting and well motivated. However, it is fairly simple. It requires point cloud inputs of the objects and then only needs to predict a binary classification if any two objects interact or not. This seems like a simplified version of "Learning Affordance Landscapes for
Interaction Exploration in 3D Environments", Nagarajan et al. 2020 which predicts actions performable on an object (e.g., open, close, toggle, etc.) from images direction (e.g., without pre-segmented objects/point clouds). Such an approach could be applied to this setting, by additionally predicting the object that is changed, by making the output also capture the relationships/changes when an action is performed. Related, this work does not compare to any previous works. While the ablations are interesting and well done, the lack of other comparisons is a weakness. Especially given that this introduces a new dataset, comparing previous works to establish baselines is helpful.

There also aren't comparisons in the accuracy of different types of relationships (toggleable, many-to-many, one-to-one, functional, inter-object, etc). It would be interesting to see which of these the model does well on and which it does not do well on.

The experiments on transferring between environments are interesting and is an important ability for such an approach.

The paper would benefit from some proofreading and editing, there are some sentences with grammar mistakes, especially in the introduction. Otherwise, the paper is well written and easy to follow and understand.

The RL and self-supervised learning approach seem effective for this problem.


**Summary Of The Paper:**

This paper proposes a method to learn relationships between two objects, e.g., flipping a switch turns a light on. This is formulated as a binary classification task taking two objects as input and outputing a single a value $\in [0,1]$ if the objects are related or not. The paper proposes a method to train this using RL to select the training (object_1, object_2) pairs. The paper also introduces a dataset for evaluation and provides a lot of experiments to study this.

**Summary Of The Review:**

Overall the paper is well motivated and the problem is interesting. However, the approach is not compared to any previous works and is fairly simple (e.g., predicting a binary value if two objects are related or not). With some revisions and additional experimental comparisons, I think the paper would be much stronger.

---

> ### Author Response · Authors · 2021-11-22
> **Response to Reviewer fQs8: Thank you so much for the constructive feedback and valuable suggestions! We’ve addressed all of your questions, added a new baseline, and hope our replies addressed your concerns. We are eager to hear back from you! (Part 1/3)**
>
> Thank you so much for the constructive feedback and valuable suggestions for further improving our work! Thank you for taking the time and effort to review our paper!
>
> We are very happy to hear that the reviewer thinks the problem is interesting and well-motivated. We are also very thankful to know that the ablations are interesting and well done, and that the experiments transferring across environments are interesting and important. We will release our code and data for the reproducibility of the work and hope to draw more research attention to this underexplored yet important task in embodied AI.
>
> **Thank you for your valuable suggestions to improve the paper! Thank you for the questions! We have addressed all of them below and revised the paper accordingly to reflect the changes (we highlight all changes in red in the revised paper)!**
>
> > The problem is fairly simple. It requires point cloud inputs of the objects and then only needs to predict a binary classification if any two objects interact or not. This seems like a simplified version of "Learning Affordance Landscapes for Interaction Exploration in 3D Environments", Nagarajan et al. 2020. which predicts actions performable on an object (e.g., open, close, toggle, etc.) from images directly (e.g., without pre-segmented objects/point clouds).
>
> Thank you for the question! Though our final output is indeed binary classification over object pairs, **the task is actually not very easy**, and there are **many non-trivial technical designs** customized for this task to work out a solution that works well. Compared to Nagarajan & Grauman (2020), **many challenges are unique to our task** and **it requires some targeted designs that actually make the system quite complicated**.
>
> We first want to clarify that **the task goals are very different**: we are solving inter-object functional relationships (IFRs) among all objects in a scene, while Nagarajan & Grauman (2020) aims for tackling a very different task on learning affordance over individual shapes in a scene and does not consider modeling relationships among objects to solve IFRs. Therefore, **there are many unique designs** to solve for the IFRs, such as modeling all IFRs in a scene graph for joint reasoning, decoupling a binary-relationship network from the scene-level one to reduce the problem complexity, performing scene-aware reasoning globally over all IFRs to figure out where to explore and interact in the next step, learning to update the IFR beliefs in the scene graph when observing new pairs of IFR during test-time interactions, and training the IFR belief updates using only partially observed ground-truth IFRs.
>
> Among these unique challenges to our task, we want to emphasize three most challenging and different points compared to Nagarajan & Grauman (2020).
>
> Firstly, **since there are intrinsic ambiguities for perceiving IFRs in a scene, we have to create a scene graph and operate on it to reason globally given all the objects in the scene.** For example, while it seems straightforward that buttons can be toggled and lamps can be turned on, as the predicted affordance in Nagarajan & Grauman (2020) may suggest, their pairings may be uncertain (e.g., which button controls which light if there are two buttons and two lights) and need to be figured out by **modeling scene-level context (e.g., a button controls the closer lamp)** or **performing strategic and efficient interactions (e.g., one needs but ONLY needs to try one pair and can automatically infer the other pair without an interaction)**. We do not need to try all IFR pairs with uncertainties but only try a few and automatically reason out the rest. To enable such reasoning capability, we model all the objects as nodes and all IFRs as edges, create a scene graph of IFRs, and design our network modules operating and reasoning over this IFR scene graph, so that IFR ambiguities can be addressed in an efficient and intelligent way. It is also non-trivial to train this smart policy as we only observe partial IFR scene graph ground-truth.
>
> Secondly, compared to employing only one RL policy in Nagarajan & Grauman (2020), we design our framework such that **the agent uses two different strategies for exploring large-scale offline scenes and adapting to a test novel scene**. Intuitively, in the training stage, the agent aims to explore large-scale offline scenes to learn as much prior knowledge as possible given the limited interaction budget and thus should focus more on exploration, while in the testing stage, the agent aims to fastly adapt to new scenes and thus should only interact with objects that the agent is uncertain about. We thus propose to use an RL agent to learn prior knowledge while utilizing the uncertainty of IFR predictions for the test-time adaptation.

---

> > ### Author Response · Authors · 2021-11-22
> > **Response to Reviewer fQs8 (Part 2/3)**
> >
> > Lastly, **the prior learning of IFRs requires many careful designs**, which is an important part of our contributions. The possible pairs of objects increase quadratically as the number of objects grows in a scene. Given a scene with 70 objects, the possible IFRs quickly grow to 4900, giving a big complexity. But, the true IFRs are very sparse (e.g., maybe only 1%). We do not want to explore all of them since it is too expensive. Thus, we need to design our networks carefully to **mine powerful semantic priors** given the input shapes and scene context. For example, our network learns not to interact with non-interesting background objects. Also, though binary-relationship prior network learns more easily as it scopes down the problem to only consider two objects, the scene-relationship networks are necessary to incorporate the scene-contextual information.
> >
> > **We hope our answers above make it clearer about the challenges of our task and our differences compared to Nagarajan & Grauman (2020). We are happy to discuss further if the reviewer has more concerns.**
> >
> > > The approach in "Learning Affordance Landscapes for Interaction Exploration in 3D Environments", Nagarajan et al. 2020” could be applied to this setting, by additionally predicting the object that is changed, by making the output also capture the relationships/changes when an action is performed. Related, this work does not compare to any previous works. While the ablations are interesting and well done, the lack of other comparisons is a weakness.
> >
> > Thank you for the nice suggestions for adding more baselines from previous works to improve our work further! We really appreciate your ideas and **have added the suggested baseline that we try our best to adapt the work Nagarajan & Grauman (2020) to our setting**. We use a PointNet++ to encode object features and an MLP to additionally predict a set of objects that each interaction at a object causes. **We add this new baseline to Sec. 5.2 and report the quantitative comparisons in Table 1 of the revised paper**, from which we see that **our method still performs the best**. We did not do this in the original submission because the tasks are quite different as detailedly discussed in the reply to the previous question.
> >
> > We have made several necessary changes to adapt Nagarajan & Grauman (2020) to our setting:
> >   -  we assume Nagarajan & Grauman (2020) have access to the pre-segmented object instances and can detect object changes after interactions, similarly to our setting. It is hard to output functional relationships/changes directly based on the original framework of Nagarajan & Grauman (2020). The responders are often out of sight (e.g., the agent can only see a light switch, without the corresponding chandelier), so we cannot model IFRs over a single image. Having pre-segmented object instances, we use a PointNet++ to extract per-object features, concatenate the per-pixel feature in the image and the per-object feature, and train an MLP to predict if the action at the pixel triggers the other objects.
> >   -  we make sure the agent uses the right interactions when interacting with the objects of interest. The IFRs could be very noisy in Nagarajan & Grauman (2020) because many irrelevant interactions exist. It would be unclear whether A and B really have no IFR or it is just because the agent picks the wrong actions (e.g., “picking up” a button won’t turn on the light).
> >
> > We think that the above changes are necessary and actually favor Nagarajan & Grauman (2020) for adapting it to our setting and using it for comparison.
> >
> > Finally, we want to clarify that, since **we are the first to study the problem** of learning inter-object functional relationship in 3D scenes, **there is no prior work that studies the same problem for us to directly and fairly compare against**. This task is very important to developing the embodied AI agents but has not been studied before, motivating us to bring people’s research attention to this problem. In addition to the added baseline, we have tried our best to conduct extensive ablation studies for quantitative comparisons and hope they also serve well as baselines on the new dataset. **Thank you so much for recognizing that our ablations are interesting and well done.**
> >
> > **We are happy to further discuss with the reviewer regarding the added experiment and would be happy to add more comparisons if suggested.**

---

> > > ### Author Response · Authors · 2021-11-22
> > > **Response to Reviewer fQs8 (Part 3/3)**
> > >
> > > > Evaluate over different types of relationships?
> > >
> > > This is a great point and is also mentioned by reviewer RFix! **We have added this and reported the performance over different types of relations** (i.e., “one-to-one”, “many-to-many”, “self relationship”, “binary relationship”) in the revised paper (see Appendix Sec. C, Table C.4).
> > >
> > > The results show that the agent actually performs very well on the “many-to-many” relationship, which is reasonable because “one-to-one” relationships would suffer more from the uncertainties and ambiguities. Another clear trend is that the agent performs much better in the “self relationship” (i.e. the object triggers itself) compared to the “binary relationship” (i.e. one object triggers another object), which is very intuitive since it is more difficult to learn functional relationships across different objects.
> > >
> > > > Do more proofreading and editing regarding grammar mistakes.
> > >
> > > Thank you for pointing this out! **We have carefully revised the paper and corrected the grammar mistakes.** We highlight our changes in red in the revised paper.
> > >
> > > **We sincerely hope that our response above makes things clearer to you and addresses your suggestions well. Otherwise, please do not hesitate to ask us more and we are very happy to discuss further. Thank you again for the useful comments and questions!**

---

> > > > ### Comment · Reviewer_fQs8 · 2021-11-30
> > > > **Response**
> > > >
> > > > Thanks for the additional details and revisions. Overall, I think the paper works on an interesting problem. While I still feel it is a simple, the response and revision has added more experiments and studies that help increase the contribution. I have increased my rating accordingly.

---

> > > > > ### Author Response · Authors · 2021-11-30
> > > > > **Thank you!**
> > > > >
> > > > > Thanks for raising your rating to 6. We are glad that our responses help alleviate your concerns. Thanks again for all your valuable feedback!

---

### Decision · Program_Chairs · 2022-01-20

**Decision:**

Accept (Poster)

**Comment:**

The paper presents an approach for learning inter-object relations. The relationships are represented in terms of scene graphs, and are processed with graph convolutional networks. All of the reviewers find the problem interesting and meaningful, which is the main strength of the paper. The approach assumes good object segmentations and state changes are provided to the agent, which the AC believes is a very dangerous assumption. Object segmentation or the change detection from raw data is still an active research area and lack of end-to-end training capability of the proposed approach is a limiting aspect. Still, the authors were able to convince all the reviewers that the problem formulation and the proposed approach is valid. Experimental results were provided to support the argument.

Respecting the opinions from the reviewers, the AC recommends accepting the paper, although the AC himself/herself is very reluctant to give an accept rating.